# PathFlow: A Normalizing Flow Generator that Finds Transition Paths

**Tianyi Liu**[1]                **Weihao Gao**[1]                **Zhirui Wang**[1]                **Chong Wang**[1]

[1]ByteDance Inc.

## Abstract

Sampling from a Boltzmann distribution to calculate important macro statistics is one of the central tasks in the study of large atomic and molecular systems. Recently, a one-shot configuration sampler, the Boltzmann generator [Noé et al., 2019], is introduced. Though a Boltzmann generator can directly generate independent metastable states, it lacks the ability to find transition pathways and describe the whole transition process. In this paper, we propose PathFlow that can function as a one-shot generator as well as a transition pathfinder. More specifically, a normalizing flow model is constructed to map the base distribution and linear interpolated path in the latent space to the Boltzmann distribution and a minimum (free) energy path in the configuration space simultaneously. PathFlow can be trained by standard gradient-based optimizers using the proposed gradient estimator with a theoretical guarantee. PathFlow, validated with the extensively studied examples including a synthetic Müller potential and Alanine dipeptide, shows a remarkable performance.

## 1   INTRODUCTION

In the study of large atomic and molecular systems, the calculation of important macro statistics such as the total energy of the system or the folding probability of a protein is of fundamental importance [Tuckerman, 2010]. One may turn to Monte Carlo methods that require unbiased sampling of the equilibrium distribution. In many applications, the distribution can be expressed by the Boltzmann distribution:

$$p(\boldsymbol{r}) = \frac{1}{Z} \exp(-\mathcal{K}(\boldsymbol{r})),$$

where $\boldsymbol{r}$ is one configuration of the system, $\mathcal{K}(\boldsymbol{r})$ represents functions depending on the potential energy of the system

e.g. the temperature and other thermodynamic quantities. The statistics are typically based on a sufficient observation of all important configurations. Whereas, the enumeration of these configurations is usually infeasible.

Recently, Noé et al. [2019] introduce a machine learning based Boltzmann distribution sampler, known as the Boltzmann generator. Following the idea of normalizing flows, Boltzmann generators seek an invertible mapping $F_{ZX}(z)$ from a latent space $Z$ to the configuration space $X$ which maps a simple Gaussian distribution to the targeted Boltzmann distribution. Unlike molecular dynamics (MD) sampling methods that require a long time simulation, Boltzmann generators can produce uncorrelated and low energy samples from different metastable states in one-shot.

Though the Boltzmann generator successfully repacks the high probability regions of the configuration space into a concentrated latent space density, its abilities to explore high energy regions and to find the transition pathways are not well justified. The synthetic experiments in Noé et al. [2019] report the feasibility of achieving transition pathways with low energy and high probabilities through mapping of the linear interpolated paths in latent space. However, there are neither theoretical results nor physical constraints to guarantee the physical meaning behind this observation. As an important concept in molecular dynamics, the transition path between metastable states provides an important description of the transition mechanism. For instance, the transition path can be used to evaluate the lowest energy barrier and the transition rate, where the rate is a good metric of materials in applications such as catalyst discovery. Meanwhile, the transition path, as an important guidance, can help to figure out the favorable condition for the transition of chemical reactions. The lack of physical interpretations of direct paths in the latent space limits the application of Boltzmann generators in transition path finding. To the best of our knowledge, however, there is no successful effort yet to improve the path finding ability of Boltzmann generators.

In this paper, an extended normalizing flow method, named

*Accepted for the 38th Conference on Uncertainty in Artificial Intelligence* (UAI 2022).

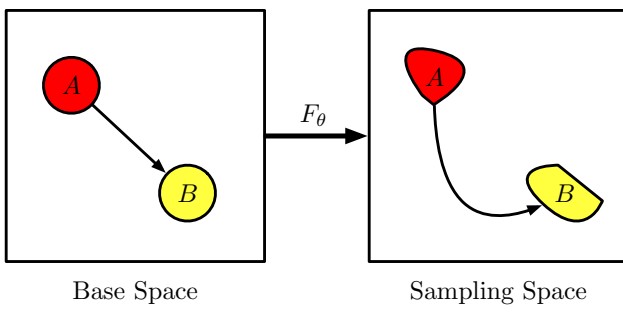

Figure 1: Illustration of PathFlow that maps the base distribution and a linear interpolated path to the Boltzmann distribution and a transition path simultaneously.

PathFlow, is proposed to improve the learning of transition paths. Beside retaining the feature of generating independent samples from the Boltzmann distribution, PathFlow further introduces physical constraints during training to regularize the mapping of linear interpolated paths between two metastable states to the *minimum energy path* (MEP) or the *minimum free energy path* (MFEP). A simple illustration of this mapping is provided in Figure 1.

Specifically, a system with two metastable states centered around $A$ and $B$ is considered. An invertible function $F$ is learnt in two modes:

*Learning on examples* follows the general training of normalizing flows where we collect data of metastable states from MD and then train the model by minimizing the negative log-likelihood loss function $L_{\mathrm{NF}}$.

*Learning on paths* is the main principle behind PathFlow. Following the physical definition of MEP and MFEP, another loss function $L_{\mathrm{path}}$ is designed to measure the ability of $F$ mapping the linear interpolated path in the latent space to a transition path with physical meaning. On-the-fly estimators of physics quantities required in the calculation of $L_{\mathrm{path}}$ as well as its gradient $\nabla L_{\mathrm{path}}$ are provided based on restraint dynamics [Maragliano et al., 2006, Maragliano and Vanden-Eijnden, 2006].

Therefore, unlike other path finding methods [Jónsson et al., 1998, Weinan et al., 2002], PathFlow can be trained by applying gradient-based methods to minimize the total loss:

$$L = w_{\mathrm{NF}}L_{\mathrm{NF}} + w_{\mathrm{path}}L_{\mathrm{path}}.$$

In the experiments based on extensively studied synthetic Müller potential and real-world Alanine dipeptide examples, a remarkable performance is achieved by PathFlow. Particularly, our contributions are summarized as below:

- Introduce physical constraints to normalizing flow which leads to a new machine learning model with knowledge of both high energy and low energy area of a system. This new model can serve as a data generator as well as a transition path finder.

- Design a loss function $L_{\mathrm{path}}$ to measure the performance of a transition path and provide its estimator based on restraint dynamics. Theoretical bounds of the estimation error are also provided.

## 2 RELATED LITERATURE

**Molecular Dynamics.** The first molecular dynamic simulations can be dated back to mid-20th century [Alder and Wainwright, 1957, McCammon et al., 1977]. Over the past several decades, with the fast development of computational sciences, MD has been successfully applied to physics, chemistry, biology, materials science, and several other fields. One of the greatest challenges of MD is to sample the rare events of state transitions. Enhanced sampling is thus needed to accelerate the dynamics. One line of research focuses on adding bias to the potential along predefined collective variables (CVs) to decrease the energy barrier. Such methods include, but are not limited to, the widely used umbrella sampling [Torrie and Valleau, 1977], adaptive biasing force method [Darve and Pohorille, 2001], metadynamics [Laio and Parrinello, 2002], and variational enhanced sampling [Valsson and Parrinello, 2014]. However, in many systems, proper CVs are not easily identified. Under such a situation, CV-free methods can be helpful. A number of such methods were proposed, such as parallel tempering [Swendsen and Wang, 1986], replica exchange of molecular dynamics [Sugita and Okamoto, 1999] and integrated tempering sampling [Gao, 2008].

**Transition Path Finding.** The study of the transition between metastable states is one of the most fundamental problems in chemistry. Existing literature such as transition state theory [Pechukas, 1981], transition path sampling [Dellago et al., 2002] and transition path theory [Vanden-Eijnden, 2006] establishes theoretical foundations to understand the mechanics of the transition. The well-known transition state theory states that the system has to navigate itself to the transition state, which is a saddle point on the potential energy surface. The most probable transition path for the reaction is the MEP. Popular methods for finding MEP include nudged elastic band (NEB) [Jónsson et al., 1998], string method [Weinan et al., 2002] and its variations [Weinan et al., 2007, Maragliano and Vanden-Eijnden, 2007, Pan et al., 2008]. Maragliano et al. [2006] extend the definition of MEP to the free energy space and modify the string method to find MFEP. After that, MFEP has been widely explored [Branduardi et al., 2007, Chen et al., 2013] and applied in different applications [Hu et al., 2007, Matsunaga et al., 2012].

**Normalizing Flow.** Normalizing flows (NF) are a family of generative models with tractable distributions where both sampling and density evaluation can be efficient and exact. It was popularised by Mohamed and Jimenez Rezende [2015] in the context of variational inference. Popular architectures include, but are not limited to, the planar flow, nonlinear

independent components estimation (NICE) [Dinh et al., 2014], real non-volume preserving (RealNVP [Dinh et al., 2017]), masked autoregressive flow (MAF, [Papamakarios et al., 2017]). Recent development on neural ordinary differential equations [Chen et al., 2018] extends discrete flow models to the continuous flow. Normalizing flows have been widely applied in different machine learning applications such as image generation [Ho et al., 2019], noise modelling [Abdelhamed et al., 2019], video generation [Kumar et al., 2019] and etc. Beside Boltzmann generators, normalizing flows also receive great attention in physics [Köhler et al., 2019, Kanwar et al., 2020, Wirnsberger et al., 2020, Wong et al., 2020, Wu et al., 2020]

## 3 MODEL

Consider a system in the NVT ensemble where the coordinates of $D$ atoms are given by $\boldsymbol{r} = (\boldsymbol{r}_1, \boldsymbol{r}_2, ..., \boldsymbol{r}_{3D}) \in \mathbb{R}^{3D}$. The potential energy of the system is denoted by $V(\boldsymbol{r})$. It is known that $\boldsymbol{r}$ follows a Boltzmann distribution:

$$p(\boldsymbol{r}) = \frac{1}{Z} \exp(-\beta V(\boldsymbol{r})),$$

where $Z = \int_{\mathbb{R}^{3D}} \exp(-\beta V(\boldsymbol{r})) d\boldsymbol{r}$ is the partition function and $\beta = \frac{1}{\kappa_\beta T}$ is the inverse temperature. Here, $\kappa_\beta$ is the Boltzmann constant and $T$ is the temperature.

Suppose the system has two metastable states $A$ and $B$, which, for instance, may represent the reactant and product states of a reaction. Based on MD simulation methods starting from $A$ and $B$, the data $\{\boldsymbol{r}_A^i\}_{i=1}^n$ and $\{\boldsymbol{r}_B^i\}_{i=1}^n$ can be sampled. However, the transition between these two states can hardly be observed without any enhanced sampling technique, because of the high energy barrier presented in the potential energy landscape. In addition, long simulation trials are always required to achieve statistically independent samples for both metastable states.

This section describes the PathFlow model, avoiding the aforementioned challenges, to generate independent metastable states samples as well as the transition path. To achieve these two goals, the model will be trained in two modes: *learning on examples* and *learning on paths*.

### 3.1 LEARNING ON EXAMPLES

Given a target distribution $X$ with probability density $p_X$, normalizing flows (NFs) target to find a learnable and invertible function $F_\theta : \mathbb{R}^d \mapsto \mathbb{R}^d$, usually represented by a neural network with parameter $\theta$, that transforms a probability density $Z$ to the target $X$, i.e., $X = F_\theta(Z)$ and $Z = F_\theta^{-1}(X)$. Allowing the change of variable rule, we know that

$$p_X(x) = p_Z(F_\theta^{-1}(x)) \left| \det(J_{F_\theta^{-1}}(x)) \right|,$$

where $J_{F_\theta^{-1}}(x)$ is the Jacobian matrix of $F_\theta^{-1}$ at $x$. Given $n$ realizations of the distribution $X$, $\{x_i\}_{i=1}^n$, NFs can be trained by minimizing the negative log-likelihood:

$$-\sum_{i=1}^n \log p_X(x_i) = -\sum_{i=1}^n \left[ \log p_Z(F_\theta^{-1}(x_i)) + \log \left| \det(J_{F_\theta^{-1}}(x_i)) \right| \right].$$

The base distribution $Z$ is usually chosen as a uni-modal Gaussian distribution or uniform distribution. However, Cornish et al. [2020] point out that NFs can hardly map a uni-modal base distribution to a multimodal distribution such as the Boltzmann distribution considered in this paper. To overcome this issue, we opt to use two separate base distributions $Z_A$ and $Z_B$ for states $A$ and $B$, respectively. Different from Boltzmann generators using two mappings for two disconnected states, we will transform the two base distributions using the same mapping $F_\theta$. We expect that:

$$Z_A = F_\theta^{-1}(\boldsymbol{r}_A) \quad \text{and} \quad Z_B = F_\theta^{-1}(\boldsymbol{r}_B).$$

The negative log-likelihood loss to find the best parameter $\theta$ can then be written as:

$$
\begin{aligned}
&L_{\text{NF}}(\theta; w_A, w_B) \\
&= w_A L_{\text{NF}}^A(\theta) + w_B L_{\text{NF}}^B(\theta) \\
&= -w_A \sum_{i=1}^n \left[ \log p_{Z_A}(F_\theta^{-1}(\boldsymbol{r}_A^i)) + \log \left| \det(J_{F_\theta^{-1}}(\boldsymbol{r}_A^i)) \right| \right] \\
&\quad - w_B \sum_{i=1}^n \left[ \log p_{Z_B}(F_\theta^{-1}(\boldsymbol{r}_B^i)) + \log \left| \det(J_{F_\theta^{-1}}(\boldsymbol{r}_B^i)) \right| \right],
\end{aligned}
\tag{1}
$$

where $(w_A, w_B)$ are the weights of the two states.

### 3.2 LEARNING ON PATHS

To enable PathFlow to find physically meaningful transition pathways, we introduce physical constraints to the model training. Here, we are especially interested in finding the minimum energy path or the minimum free energy path.

#### 3.2.1 Minimum Energy Path (MEP)

An MEP is a path that connects two minima of $V(\boldsymbol{r})$ via a saddle point and corresponds to the steepest descent path on $V(\boldsymbol{r})$ from this saddle point. More specifically, each point on the MEP is a local potential energy minimum on the hyperplane tangent to the path. This implies that the force $-\nabla V$ must be everywhere tangent to the MEP. Denote the MEP by a curve $\boldsymbol{r}(\alpha)$, where $\alpha \in [0, 1]$ is a parametrization of the path. We then have, for $\forall \alpha \in [0, 1]$,

$$\nabla V(\boldsymbol{r}(\alpha)) \text{ is parallel to } \frac{d\boldsymbol{r}(\alpha)}{d\alpha}, \tag{2}$$

or equivalently,

$$\nabla V(\boldsymbol{r}(\alpha)) - (\nabla V(\boldsymbol{r}(\alpha)) \cdot \hat{t})\hat{t} = 0, \qquad (3)$$

where $\hat{t}$ is the unit tangent vector along the path at $\boldsymbol{r}(\alpha)$. Eq. (3) is not yet a numerically efficient way to measure the performance of a path, due to the high computational cost to calculate the tangent vector. Olender and Elber [1997] instead prove that finding MEP is equivalent to solving the following variation optimization problem:

$$P_{\text{MEP}} = \underset{P:A \to B}{\operatorname{argmin}} \int_P \|\nabla V\|_2 |dl|, \qquad (4)$$

for a gradient system

$$\frac{d\boldsymbol{r}}{d\alpha} = -\nabla V(\boldsymbol{r}(\alpha)).$$

Suppose the path is divided into $S$ segments $\{l_i\}_{i=1}^S$ with arc lengths $\{dl_i\}_{i=1}^S$. Let $-\nabla V_i$ be the force at the starting point of $i$-th path segment. The discretization of the optimization objective in Eq. (4) provides us an ideal loss function to measure the performance of a candidate path $P$:

$$L_{\text{MEP}}(P) = \sum_{i=1}^S \|\nabla V_i\|_2 |dl_i|. \qquad (5)$$

### 3.2.2 Minimum Free Energy Path (MFEP)

Finding MEP has to deal with the difficulty caused by the extremely high dimensionality of the system and also the non-smoothness of the potential energy landscape. This difficulty can be reduced by the introduction of collective variables (CVs) and the mapping of MEP to the CV space (denoted as $\mathcal{X}$). Given $N$ predefined CVs denoted by $\boldsymbol{x}(\boldsymbol{r}) = (x_1(\boldsymbol{r}), ..., x_N(\boldsymbol{r}))$, the free energy associated with $\boldsymbol{x}(\boldsymbol{r})$ is defined as follows:

$$U(\boldsymbol{z}) = -\beta^{-1} \ln \left( Z^{-1} \int_{\mathbb{R}^{3D}} e^{-\beta V(\boldsymbol{r})} \right.$$
$$\left. \times \prod_{i=1}^N \delta(x_i(\boldsymbol{r}) - z_i)d\boldsymbol{r} \right), \forall \boldsymbol{z} \in \mathcal{X}, \qquad (6)$$

where $\delta$ is the Dirac delta function. On the free energy surface, the path of our interest is the minimum free energy path (MFEP). Letting $\boldsymbol{z}(\alpha) = \boldsymbol{x}(\boldsymbol{r}(\alpha))$, Maragliano et al. [2006] show that MFEP $\boldsymbol{z}(\alpha)$ must satisfy

$$\frac{d\boldsymbol{z}(\alpha)}{d\alpha} \text{ is parallel to } M(\boldsymbol{z}(\alpha))\nabla_{\boldsymbol{z}} U(\boldsymbol{z}(\alpha)), \qquad (7)$$

where

$$M_{ij}(\boldsymbol{z}) = Z^{-1} e^{\beta U(\boldsymbol{x})} \int_{\mathbb{R}^{3D}} \sum_k \frac{\partial x_i(\boldsymbol{r}(\alpha))}{\partial r_k} \frac{\partial x_j(\boldsymbol{r}(\alpha))}{\partial r_k}$$
$$e^{-\beta V(\boldsymbol{r})} \prod_{i=1}^N (z_i - x_i(\boldsymbol{r}))d\boldsymbol{r}. \qquad (8)$$

Maragliano et al. [2006] also prove that MFEP is the most likely path of transitions between $A$ and $B$. Hence, it can greatly help us understand the underlying physical mechanism of the transition. Similar to Eq. (5), the following loss can be utilized to measure the performance of a candidate path on the free energy surface.

$$L_{\text{MFEP}}(P) = \sum_{i=1}^S \|M_i \nabla U_i\|_2 |dl_i|, \qquad (9)$$

where $P$ is a candidate path in $\mathcal{X}$ connecting A and B.

*Remark* 3.1. The minimum energy path can be viewed as a special case of the minimum free energy path. Specifically, if we choose $\boldsymbol{x}(\boldsymbol{r}) = \boldsymbol{r}$, i.e., an identity mapping, the free energy $U$ is exactly the potential energy $V$ and the transition matrix as defined in Eq. (8) is reduced to an identity matrix. Therefore, Eq. (9) is the same as Eq. (5).

### 3.3 TOTAL LOSS DESIGN

It is necessary to find an invertible mapping $F_\theta$ that 1) maps the base distribution to the target Boltzmann distribution and 2) maps a base path in the latent space to a transition path in the configuration or CV spaces. Given a base path $P_{\text{base}}$, the path after mapping is denoted as $F_\theta(P_{\text{base}})$. We can then use Eqs. (1), (5) and (9) to measure how good the parameter $\theta$ is to realize two targets. Denote the path loss as

$$L_{\text{path}}(\theta) = \begin{cases} L_{\text{MEP}}(F_\theta(P_{\text{base}})), \text{ or} \\ L_{\text{MFEP}}(F_\theta(P_{\text{base}})). \end{cases} \qquad (10)$$

Combining the path loss Eq. (10) with NF loss Eq. (1), we obtain the loss to train PathFlow:

$$L(\theta) = w_{\text{NF}} L_{\text{NF}}(\theta) + w_{\text{path}} L_{\text{path}}(\theta), \qquad (11)$$

where $w_{\text{NF}}$ and $w_{\text{path}}$ are two hyper parameters to control the weight of two losses. To ease the training of our model, the base path $P_{\text{base}}$ and base distribution $Z_A, Z_B$ should be carefully selected. First, the end-points $A$ and $B$ of the transition path need to be determined. In some applications like the study of chemical reactions, the start and end of the transition path is already known. In other cases, $A$ and $B$ can be set in terms of the simulation data. For example, the end-points can be chosen as

$$\boldsymbol{\mu}_A, \boldsymbol{\mu}_B = \begin{cases} \frac{1}{n}\sum_i \boldsymbol{r}_A^i, \frac{1}{n}\sum_i \boldsymbol{r}_B^i, \text{ if in configuration space;} \\ \frac{1}{n}\sum_i \boldsymbol{x}(\boldsymbol{r}_A^i), \frac{1}{n}\sum_i \boldsymbol{x}(\boldsymbol{r}_B^i), \text{ if in CV space,} \end{cases}$$
$$(12)$$

the mean of samples from states A and B, respectively. Since the transition path must have $A$ and $B$ as its end-points, we require the base path starting from $F_\theta^{-1}(A)$ and ending at $F_\theta^{-1}(B)$. A natural choice of the whole base path is the linear interpolated path between these two points, i.e.,

$$P_{\text{base}}(\alpha) = (1 - \alpha)F_\theta^{-1}(A) + \alpha F_\theta^{-1}(B).$$

In the sampling space, the simulation data should center around the points with minimal potential or free energy, which at the same time are the end-points of the transition path. Therefore, we prefer to set

$$Z_A \sim Gaussian(F_\theta^{-1}(A), \sigma \boldsymbol{I}),$$
$$Z_B \sim Gaussian(F_\theta^{-1}(B), \sigma \boldsymbol{I}).$$

Here, $\sigma$ can be used to control the concentration of the base distribution. When $\sigma$ is small, most of the linear interpolated path lies outside the concentration area of $Z_A$ and $Z_B$. Hence, the model can focus on learning the path only. On the other hand, co-training of the path and the generator may be difficult around A and B, since $F_\theta$ has to minimize two losses at the same time. However, since we are most interested in the transition process that happens around the high energy barrier, we can avoid this conflict by reducing the weight of path samples near the end-points.

## 3.4 GRADIENT-BASED TRAINING

The gradient descent type algorithm is applied to update the model parameter $\theta$ to minimize $L(\theta)$. Notice that

$$\nabla_\theta L(\theta) = w_{\text{NF}} \nabla_\theta L_{\text{NF}}(\theta) + w_{\text{path}} \nabla_\theta L_{\text{path}}(\theta).$$

The gradient $\nabla_\theta L_{\text{NF}}(\theta)$ can be calculated by backpropagation that has already been implemented in popular deep learning frameworks such as Tensorflow [Abadi et al., 2016] and PyTorch [Paszke et al., 2019]. $\nabla_\theta L_{\text{path}}(\theta)$, however, involves the calculation of the potential mean force, the transition matrix and their gradients, and therefore cannot be calculated automatically. In the next section, we will provide efficient estimators of all the physics quantities appearing in $\nabla_\theta L_{\text{path}}(\theta)$ using restraint dynamics.

## 4 GRADIENT ESTIMATOR BY RESTRAINT DYNAMICS

In this section, we provide an estimator of the gradient $\nabla_\theta L_{\text{path}}(\theta)$ to facilitate the gradient-based training of our model. Since MEP can be viewed as a special case of MFEP as we mentioned in Remark 3.1, we only consider the case where $L_{\text{path}} = L_{\text{MFEP}}$. Suppose, under parameter $\theta$, the candidate path is $P(\theta)$ and of arc length $l(P(\theta))$. We uniformly divide $P(\theta)$ into $S$ segments, each with an arc length $|dl_i| = l(P(\theta))/S$ in Eq. (9). We then have:

$$L_{\text{MFEP}}(P) = l(P(\theta)) \frac{1}{S} \sum_{i=1}^{S} \|M_i \nabla U_i\|_2,$$

as well as the gradient:

$$\nabla_\theta L_{\text{path}}(\theta) = \nabla_\theta l(P(\theta)) \frac{1}{S} \sum_{i=1}^{S} \|M_i \nabla U_i\|_2$$
$$+ l(P(\theta)) \frac{1}{S} \sum_{i=1}^{S} \nabla_\theta \|M_i \nabla U_i\|_2. \quad (13)$$

The calculation of $\|M_i \nabla U_i\|_2$ and its gradient $\nabla_\theta \|M_i \nabla U_i\|_2$ can be done by regular backpropagation only if the free energy surface (FES) $U$ or its analytical approximation are known. In practice, however, establishing FES requires a large number of simulations, which can be a task even harder than finding the path. Therefore, it would be more efficient to estimate these values on the fly at the given sample points on $P(\theta)$. We will adopt the approach of restrained dynamics [Maragliano et al., 2006, Maragliano and Vanden-Eijnden, 2006]. For a given point $\boldsymbol{z} = (z_1, ..., z_M)$ in the CV space, this method adds a harmonic restraint to the potential of the system to represent the effect of the spring forces between the configuration variables and the CVs:

$$V_k(\boldsymbol{r}; \boldsymbol{z}) = V(\boldsymbol{r}) + \frac{k}{2} \sum_{i=1}^{N} (x_i(\boldsymbol{r}) - z_i)^2, \quad (14)$$

where $k$ is a parameter to control the restraint. The movement of particles in the CV space under this extended potential can then be characterized by the overdamped Langevin dynamics:

$$\dot{\boldsymbol{r}}(t) = -\nabla V_k(\boldsymbol{r}(t), \boldsymbol{z}) + \sqrt{2\kappa_\beta T} \eta_t, \quad (15)$$

where $\eta(t)$ is a white Gaussian noise with unit variance. It can been shown that Eq. (15) has the following Boltzmann-Gibbs density as its stationary distribution:

$$p_k(\boldsymbol{r}; \boldsymbol{z}) = \frac{1}{Z_k(\boldsymbol{z})} \exp(-\beta V_k(\boldsymbol{r}; \boldsymbol{z})),$$

where $Z_k(\boldsymbol{z}) = \int \exp(-\beta V_k(\boldsymbol{r}, \boldsymbol{z})) d\boldsymbol{r}$.

**Estimation of** $\|M \nabla U\|_2$. Define the effective free energy corresponding to $V_k(\boldsymbol{r}; \boldsymbol{z})$ as

$$U^{(k)}(\boldsymbol{z}) = -\beta^{-1} \ln \left( Z^{-1} \int_{\mathbb{R}^{3D}} \exp(-\beta V_k(\boldsymbol{r}; \boldsymbol{z})) d\boldsymbol{r} \right).$$

Maragliano et al. [2006] prove that when $k$ is large,

$$\lim_{k \to \infty} \nabla U^{(k)}(\boldsymbol{z}) = \nabla U(\boldsymbol{z}),$$

where

$$\nabla_i U^{(k)} = \int_{\mathbb{R}^{3D}} k(z_i - x_i(\boldsymbol{r})) p_k(\boldsymbol{r}; \boldsymbol{z}) d\boldsymbol{r}, \quad \forall i \leq N.$$

If we further assume the ergodicity of dynamics Eq. (15), we can obtain an estimator of the potential mean force:

$$\nabla_i U^{(T,k)}(\boldsymbol{z}) = \frac{k}{T} \int_0^T (z_i - x_i(\boldsymbol{r}(t))) dt. \quad (16)$$

Similar analysis can be done on $M$, and an estimator of $M_{ij}(\boldsymbol{z})$ can be derived from Eq. (8) as follows:

$$M_{ij}^{(T,k)}(\boldsymbol{z}) = \frac{1}{T} \int_0^T \sum_k \frac{\partial x_i(r(t))}{\partial r_k} \frac{\partial x_j(r(t))}{\partial r_k} dt. \quad (17)$$

Combining Eqs. (16) and (17), we obtain an estimation of $\|M\nabla U\|_2$ as $\|\nabla U^{(T,k)} M^{(T,k)}\|_2$.

**Estimation of** $\nabla_\theta \|M\nabla U\|_2$**.** To estimate $\nabla_\theta \|M\nabla U\|_2$ in the second term of Eq. (13), one naive approach is to use the finite difference method requiring at least $O(N)$ simulation trails under Eq. (15), which may be computationally challenged in practice. Instead, we propose a new estimator that can be obtained simultaneously with Eqs. (16) and (17). Specifically, we rewrite the gradient of $\|M\nabla U\|_2$ as follows:

$$\nabla_\theta \|M\nabla U\|_2 = \frac{J(M\nabla U)^\top M\nabla U}{\|M\nabla U\|_2},$$

where $J(\cdot)$ is the Jacobian matrix of a given function. Note that the Jacobian matrix can be further decomposed as

$$J(M\nabla U) = \nabla M\nabla U + M\nabla^2 U, \quad (18)$$

where $\nabla M\nabla U = [\nabla_{z_1} M\nabla U, ..., \nabla_{z_N} M\nabla U]$. Recall that the estimators Eqs. (16) and (17) can all be viewed as a time average estimation of the expectation of a function $f(\boldsymbol{r}, \boldsymbol{z})$ over distribution $p_k(\boldsymbol{r}; \boldsymbol{z})$, i.e., $\int_{\mathbb{R}^{3D}} f(\boldsymbol{r}, \boldsymbol{z}) p_k(\boldsymbol{r}; \boldsymbol{z}) d\boldsymbol{r}$. Specifically, for $M_{ij}$, $f(\boldsymbol{r}, \boldsymbol{z})$ is taken as $\sum_k \frac{\partial x_i(r)}{\partial r_k} \frac{\partial x_j(r)}{\partial r_k}$ and for $\nabla_i U(\boldsymbol{z})$, $f(\boldsymbol{r}, \boldsymbol{z})$ is taken as $k(z_i - x_i(\boldsymbol{r}))$. For this expectation, Maragliano et al. [2006] have proved that

$$\lim_{k \to \infty} \int_{\mathbb{R}^{3D}} f(\boldsymbol{r}, \boldsymbol{z}) p_k(\boldsymbol{r}; \boldsymbol{z}) d\boldsymbol{r}$$
$$= Z^{-1} e^{\beta U(\boldsymbol{z})} \int_{\mathbb{R}^{3D}} f(\boldsymbol{r}, \boldsymbol{z}) e^{-\beta V(\boldsymbol{r})} \prod_{i=1}^N \delta(z_i - x_i(\boldsymbol{r})) d\boldsymbol{r}.$$

Under certain regularity conditions that we can change the order of derivative and limit, as well as the order of derivative and integration, the following equation is established.

$$\lim_{k \to \infty} \int_{\mathbb{R}^{3D}} \frac{\partial f(\boldsymbol{r}, \boldsymbol{z}) p_k(\boldsymbol{r}; \boldsymbol{z})}{\partial z_l} d\boldsymbol{r}$$
$$= \lim_{k \to \infty} \frac{\partial \int_{\mathbb{R}^{3D}} f(\boldsymbol{r}, \boldsymbol{z}) p_k(\boldsymbol{r}; \boldsymbol{z}) d\boldsymbol{r}}{\partial z_l}$$
$$= \frac{\partial Z^{-1} e^{\beta U(\boldsymbol{z})} \int_{\mathbb{R}^{3D}} f(\boldsymbol{r}, \boldsymbol{z}) e^{-\beta V(\boldsymbol{r})} \prod_{i=1}^N \delta(z_i - x_i(\boldsymbol{r})) d\boldsymbol{r}}{\partial z_l}.$$

Estimating $\frac{\partial M}{\partial z_l}$ and $\nabla^2 U$ can both be generalized as how to use the simulation trajectory $\boldsymbol{r}(t)$ to estimate $\int_{\mathbb{R}^{3D}} \frac{\partial f(\boldsymbol{r}, \boldsymbol{z}) p_k(\boldsymbol{r}; \boldsymbol{z})}{\partial z_l} d\boldsymbol{r}$. By some manipulation, we have

$$\int_{\mathbb{R}^{3D}} \frac{\partial f(\boldsymbol{r}, \boldsymbol{z}) p_k(\boldsymbol{r}; \boldsymbol{z})}{\partial z_l} d\boldsymbol{r} = \int_{\mathbb{R}^{3D}} \frac{\partial f(\boldsymbol{r}, \boldsymbol{z})}{\partial z_l} p_k(\boldsymbol{r}; \boldsymbol{z}) d\boldsymbol{r}$$
$$+ \int_{\mathbb{R}^{3D}} f(\boldsymbol{r}, \boldsymbol{z}) \beta k(x_l(\boldsymbol{r}) - z_l) p_k(\boldsymbol{r}; \boldsymbol{z}) d\boldsymbol{r}$$
$$- \int_{\mathbb{R}^{3D}} f(\boldsymbol{r}, \boldsymbol{z}) p_k(\boldsymbol{r}; \boldsymbol{z}) d\boldsymbol{r} \int_{\mathbb{R}^{3D}} \beta k(x_l(\boldsymbol{r}) - z_l) p_k(\boldsymbol{r}; \boldsymbol{z}) d\boldsymbol{r}.$$

All terms are expectations under density $p_k(\boldsymbol{r}; \boldsymbol{z})$. Therefore, with ergodicity, we can use time average to construct the estimator:

$$\int_{\mathbb{R}^{3D}} \frac{\partial f(\boldsymbol{r}, \boldsymbol{z}) p_k(\boldsymbol{r}; \boldsymbol{z})}{\partial z_l} d\boldsymbol{r} \approx \frac{1}{T} \int_{t=0}^T \frac{\partial f(\boldsymbol{r}(t), \boldsymbol{z})}{\partial z_l} dt$$
$$+ \frac{1}{T} \int_{t=0}^T f(\boldsymbol{r}(t)) \beta k(x_l(\boldsymbol{r}(t)) - z_l) dt$$
$$- \frac{1}{T} \int_{t=0}^T f(\boldsymbol{r}(t), z) dt \frac{1}{T} \int_{t=0}^T \beta k(x_l(\boldsymbol{r}(t)) - z_l) dt$$
$$\triangleq \mathcal{F}_l(f(\boldsymbol{r}, \boldsymbol{z}), T, k). \quad (19)$$

Plugging $\sum_k \frac{\partial x_i(\boldsymbol{r})}{\partial r_k} \frac{\partial x_j(\boldsymbol{r})}{\partial r_k}$ or $k(z_j - x_j(\boldsymbol{r}))$ into $f(\boldsymbol{r}, \boldsymbol{z})$, we get the estimators of $\nabla_l M_{i,j}(\boldsymbol{z})$ and $\nabla_{i,j}^2 U(\boldsymbol{z})$.

$$\nabla_l M_{ij}^{(T,k)}(\boldsymbol{z}) = \mathcal{F}_l\left(\sum_k \frac{\partial x_i(\boldsymbol{r})}{\partial r_k} \frac{\partial x_j(\boldsymbol{r})}{\partial r_k}, T, k\right),$$
$$\nabla_{i,j}^2 U^{(T,k)}(\boldsymbol{z}) = \mathcal{F}_i\left(k(z_j - x_j(\boldsymbol{r})), T, k\right). \quad (20)$$

Using Eqs. (16), (17) and (20), the approximation of the Jacobian matrix in Eq. (18) is established. We are ready to use gradient-based algorithm to find $\theta$ that optimizes $L(\theta)$.

## 4.1 ESTIMATION ERROR

The following theorem shows the estimation error of estimators Eqs. (16), (17) and (20).

**Theorem 4.1.** *Suppose the dynamics Eq. (15) is ergodic, for $\forall i, j, l \leq N$ and $\boldsymbol{z}$ in $\mathcal{X}$, the estimation errors of $M_{ij}^{(T,k)}(\boldsymbol{z}), \nabla_i U^{(T,k)}(\boldsymbol{z}), \nabla_l M_{ij}^{(T,k)}(\boldsymbol{z}), \nabla_{ij}^2 U^{(T,k)}(\boldsymbol{z})$ are as follows.*

$$|M_{ij}^{(T,k)}(\boldsymbol{z}) - M_{ij}(\boldsymbol{z})| \leq O(\frac{1}{k}) + O(\frac{1}{\sqrt{T}}),$$
$$|\nabla_i U^{(T,k)}(\boldsymbol{z}) - \nabla_i U(\boldsymbol{z})| \leq O(\frac{1}{k}) + O(\frac{k}{\sqrt{T}}),$$
$$|\nabla_l M_{ij}^{(T,k)}(\boldsymbol{z}) - \nabla_l M_{ij}(\boldsymbol{z})| \leq O(\frac{1}{k}) + O(\frac{k}{\sqrt{T}}),$$
$$|\nabla_{ij}^2 U^{(T,k)}(\boldsymbol{z}) - \nabla_{ij}^2 U(\boldsymbol{z})| \leq O(\frac{1}{k}) + O(\frac{k^2}{\sqrt{T}}).$$

The proof of Theorem 4.1 can be found in Appendix A. To achieve an error of order $\epsilon$, $M_{ij}^{(T,k)}(\boldsymbol{z})$, $\nabla_i U^{(T,k)}(\boldsymbol{z})$

and $\nabla_l M_{ij}^{(T,k)}(z)$ require at most $T = O(1/\epsilon^4)$, while $\nabla_{ij}^2 U^{(T,k)}(z)$ requires $T = O(1/\epsilon^6)$. This is consistent with our empirical observation that using $\nabla_{ij}^2 U^{(T,k)}(z)$ to estimate $\nabla^2 U$ can be statistically unstable which leads to the high variance of the whole Jacobian matrix estimation.

To overcome this issue, we propose a method that uses one more simulation trial to avoid estimation of $\nabla^2 U$. Note that by Eq. (18), $\nabla_\theta \|M\nabla U\|_2$ can be decomposed as

$$\nabla_\theta \|M\nabla U\|_2 = \frac{(\nabla M\nabla U)^\top M\nabla U}{\|M\nabla U\|_2} + \nabla^2 U \frac{M^\top M\nabla U}{\|M\nabla U\|_2}.$$

The second order term $\nabla^2 U$ appears in the second term in the form of a Hessian-vector product, which can be estimated directly with one additional simulation trial independently of $N$. Specifically, let $v = \frac{M^\top M\nabla U}{\|M\nabla U\|_2}$ and we have:

$$\nabla^2 U v \approx \frac{\nabla U(z + \delta v) - \nabla U(z)}{\delta}.$$

Only one extra restraint simulation centered at $z + \delta v$ is required to get the estimate. Moreover, to increase stability, the product can also be estimated by central difference.

$$\nabla^2 U v \approx \frac{\nabla U(z + \delta v) - \nabla U(z - \delta v)}{2\delta},$$

By using Hessian-vector product trick, we obtain a new estimation of the second term.

$$\nabla^2 U \frac{M^\top M\nabla U}{\|M\nabla U\|_2} \approx \frac{\nabla U^{(T,k)}(z + \delta v^{(T,k)}(z))}{2\delta} - \frac{\nabla U^{(T,k)}(z - \delta v^{(T,k)}(z))}{2\delta},$$

where $v^{(T,k)}(z) = \frac{(M^{(T,k)})^\top M^{(T,k)} \nabla U^{(T,k)}(z)}{\|M^{(T,k)} \nabla U^{(T,k)}(z)\|_2}$. Empirically, we find that using this trick can greatly stabilize the estimation with an acceptable simulation budget increment. For more detailed error estimation, please refer to Appendix B.

## 5 NUMERICAL EXAMPLE: MÜLLER POTENTIAL

We first illustrate PathFlow using a two-dimensional Müller potential that has metastable states separated by high energy barriers. The Müller potential has an explicit formulation:

$$V(x,y) = \sum_{k=1}^{4} A_k e^{B_k}, \tag{21}$$

where we take

$$B_k = a_k(x - x_k^0)^2 + b_k(x - x_k^0)(y - y_k^0) + c_k(y - y_k^0)^2.$$

Values of all parameters can be found in Appendix C. The two metastable states of Müller potential are located around

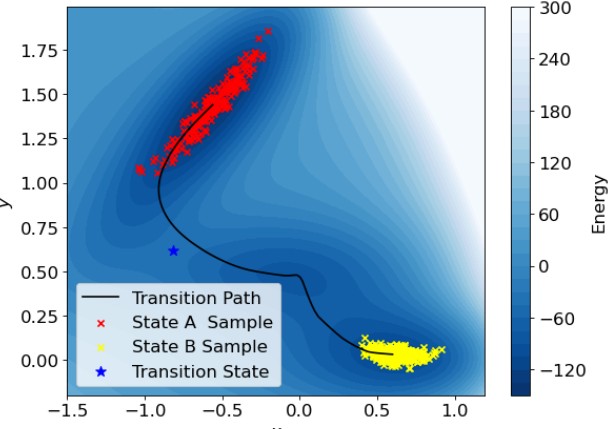

Figure 2: Experiment result on Müller Potential. PathFlow generates samples filling in two low energy regions. At the same time, the transition path found passes near the transition state. The energy barrier we found has energy of $-38$ which is very close to the ground-truth value $-40$.

$A = [-0.56, 1.44]$ and $B = [-0.05, 0.47]$, while the transition state is located around $C = [-0.82, 0.62]$. For simplicity, we consider finding the minimum energy path (MEP) starting from state A and ending at state B. We collect 100 data points using Markov Chain Monte Carlo starting from A and B respectively for learning on examples. Our normalizing flow is a masked autoregressive flow (MAF) model with 10 autoregressive layers and hidden units of shape $[256, 128, 64]$ with ReLU activation.

Given the explicit formulation of $V(x,y)$, there is no need of estimating the gradient of $L(\theta)$ using the proposed method in Section 4. All the gradients can be automatically obtained by backpropagation implemented in Tensorflow 2.3. We train the model by Adam optimizer. As shown in figure 2, PathFlow can learn the transition path and the sampler of metastable states at the same time. 1) In terms of path finding, PathFlow finds a transition path that passes the transition state $C$. The optimal energy barrier has energy around -40. The energy barrier we found is around -38 which is very close to the ground-truth. 2) In terms of sample generator, we can successfully generate data points for metastable states in one-shot.

## 6 NUMERICAL EXAMPLE: ALANINE DIPEPTIDE

In this section, we provide a practical example to illustrate the performance of our proposed models.

We study the isomerization transition and sampling of Alanine dipeptide modeled by the CHARMM27 force field [Brooks et al., 2009] at 300 K in vacuum. This transition happens between two metastable states named $C_{7eq}$

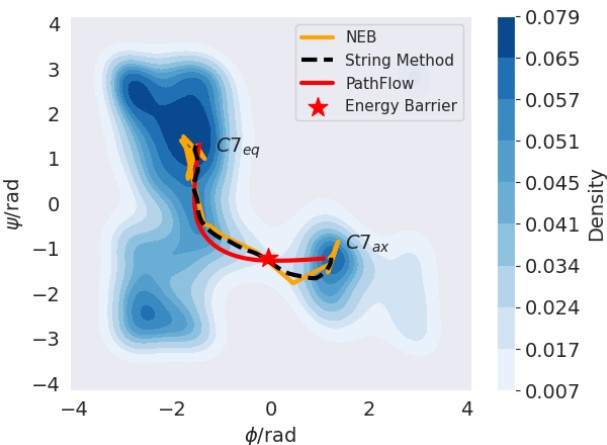

Figure 3: Experiment result on Alanine dipeptide in vacuum under room temperature 300K. The under-layer density plot is the kernel density estimation of the Boltzmann Distribution generated by Meta Dynamics. Transition pathways found by PathFlow, string method and NEB overlap in most regions. The energy barrier with the energy of about 8.6 kcal/mol lies on all paths.

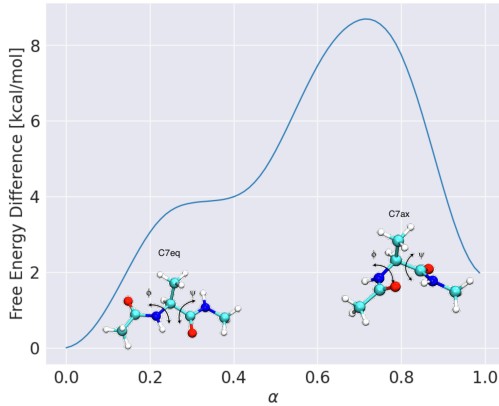

Figure 4: Free energy profile of the transition pathway found by PathFlow. Free energy in $C_{7eq}$ ($\alpha = 0$) is set as 0. The configuration plots were made by Cuny et al. [2017].

|  | $C7_{eq}$ | $C7_{ax}$ | Average |
|---|---|---|---|
| Boltzmann | -0.3889 | 1.689 | 0.6498 |
| PathFlow | -1.005 | 0.1581 | -0.4235 |
| BG Separate | -1.097 | 0.03027 | -0.5333 |

Table 1: Test Negative Log Likelihood of PathFlow, Boltzmann Generator and BG Separate.

and $C_{7ax}$. We choose two torsion angles $\phi(C, N, C_\alpha, C)$ and $\psi(N, C_\alpha, C, N)$ as our CVs for this system, i.e., $\boldsymbol{z} = (\phi, \psi)$. All the MD simulations are performed by the package GROMACS 2021 [Lindahl et al., 2021] linked with Plumed 2.7 [Tribello et al., 2014]. To generate data in two metastable states, we run brute-force MD simulations starting from $C_{7eq}$ and $C_{7ax}$ for 100 picoseconds (ps), respectively. The CV values along the MD trajectories are computed and recorded in every 0.2 ps. We randomly select 100 data points for each state to train the sampler. On each candidate path in the CV space, we sample a point every 0.1 arc length. For each sample on the path, we run three restraint simulations with $k = 500$ kJ/mol/rad for 100 ps. The CV values along the trajectories are computed and recorded in every 0.01 ps to estimate the potential mean force, transformation matrix $M$, and their derivative. We choose a masked autoregressive flow with 15 autoregressive layers and hidden units of shape $[512, 256, 128, 64]$ with ELU activation as our normalizing flow model.

**Path Finding.** To illustrate the path-finding ability of Path-Flow, we compare our model with Nudged Elastic Band (NEB) and the string method with swarms of trajectory. All the methods are implemented with 40 images. The detailed setting up of the string method follows that in Pan et al. [2008] Section III.1.

Figure 3 plots the transition pathways found by NEB (average of 30-40 iterations), the string method (average of 60-70 iterations) and PathFlow. We observe that transition paths found by PathFlow, NEB and the string method overlap in most regions. They all pass the same energy barrier with free energy difference of 8.6 kcal/mol. The three pathways

differ around $C_{7ax}$ which may be caused by the conflict between $L_{NF}$ and $L_{path}$ during training. However, the free energy profile of our pathway in Figure 4 is almost consistent with that of the string method in Pan et al. [2008].

**Configuration Generation.** We also compare PathFlow with the Boltzmann generator on Alanine Dipeptide configurations. The Boltzmann generator is trained using a Gaussian base distribution and simulation samples from both state $C7_{eq}$ and $C7_{ax}$. We expect that the Boltzmann generator is not effective at sampling separated and disconnected states, and hence we further trained two separate Boltzmann Generators (BG Separate)for these two states, respectively. We tested three models on 100 samples from each state. The test negative log likelihood is listed in Table 1.

We observe that BG Separate performs well on both states, but the Boltzmann generator achieves the worst test loss among all models. This confirms that the Boltzmann generator is not effective at sampling multi-modal distributions with two metastable states, which is widely known as a major challenge for generative models. However, by introducing two base distributions, our model PathFlow outperforms Boltzmann generators significantly in sampling multi-modal distributions. PathFlow obtains a test loss close to that of BG separate but only uses half the model size.

# 7 CONCLUSION AND PERSPECTIVE

In summary, PathFlow is a promising tool for generating Boltzmann samples and discovering transition paths to describe the transition mechanisms. Different from existing path finding algorithms (e.g.,NEB [Jónsson et al., 1998], string method [Weinan et al., 2002]), PathFlow is trained by the standard gradient-based optimizers associating with the efficient gradient estimator developed in section 4. Note that the estimator has the potential to be employed by other machine learning based path finding algorithms. In particular, as an independent research interest, it is empirically found that the gradient-based training leads to a faster path finding speed and fewer simulation trials. In addition, PathFlow can be viewed as one successful application of multitask learning to physics. We expect more multitask learning techniques will demonstrate their power in scientific research. Future research directions also include normalizing flows or other machine learning based methods in the transition tube [Vanden-Eijnden, 2006] sampling as well as CV discovery.

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
