# OpenReview forum: "PathFlow: A Normalizing Flow Generator that Finds Transition Paths"
_auai.org/UAI/2022/Conference — UAI 2022 Poster_

### Official Review · Reviewer_xtSf · 2022-04-09

**Q2(1) Originality/Novelty:** 3
**Q2(2) Significance/Impact:** 3
**Q2(3) Correctness/Technical Quality:** 3
**Q2(6) Clarity Of Writing:** 3
**Q6 Overall Score:** 7
**Q8 Confidence In Your Score:** 3

**Q1 Summary And Contributions:**

This paper studies sampling from Boltzmann distribution through a normalizing flow based generator. They propose to not only generate samples from metastable states but also find the transition pathways between states. Specifically, they use a flow model to map the linear interpolated path in latent space to the minimum (free) energy path in the configuration space.

**Q2 Assessment Of The Paper:**

More detailed information regarding each of these aspects is given below:

**Q2(4) Quality Of Experiments (Optional):**

3: Good: The experimental evaluation is adequate, and the results convincingly support the main claims.

**Q2(5) Reproducibility:**

2: Fair: Key resources (e.g., proofs, code, data) are unavailable but key details (e.g., proof sketches, experimental setup) are sufficiently well-described for an expert to confidently reproduce the main results.

**Q3 Main Strengths:**

1. The proposed method extends the Boltzmann generator to find the transition pathways between states, which is novel and potentially has many useful applications.
2. The authors propose gradient estimators for both the  minimum energy path and minimum free energy path.
3. Experiments on real-world application scenario verifies the ability to find the pathways

**Q4 Main Weakness:**

1. This might be a dumb question, but I notice all experiments deal with dynamics with two states. What if there is multiple states and we want to find pathways between any two of them? Do we need to train different models for each combination?

**Q5 Detailed Comments To The Authors:**

The paper is well written. I'm not familiar enough to this field to find missing literatures.

**Q7 Justification For Your Score:**

The proposed method is sound and novel.

**Q9 Complying With Reviewing Instructions:**

1: Yes.

---

### Official Review · Reviewer_g8Zi · 2022-04-10

**Q2(1) Originality/Novelty:** 3
**Q2(2) Significance/Impact:** 2
**Q2(3) Correctness/Technical Quality:** 3
**Q2(6) Clarity Of Writing:** 3
**Q6 Overall Score:** 5
**Q8 Confidence In Your Score:** 3

**Q1 Summary And Contributions:**

This paper presents a novel method for generating Boltzmann samples in order to discover transition paths. The key contributions of this work are well presented.


**Q2 Assessment Of The Paper:**

More detailed information regarding each of these aspects is given below:

**Q2(4) Quality Of Experiments (Optional):**

2: Fair: The experimental evaluation is weak: important baselines are missing, or the results do not adequately support the main claims.

**Q2(5) Reproducibility:**

3: Good: Key resources (e.g., proofs, code, data) are available and key details (e.g., proofs, experimental setup) are sufficiently well-described for competent researchers to confidently reproduce the main results.

**Q3 Main Strengths:**

A main strength of the paper is the interest in proposing a normalizing flow that seeks to discover transition mechanisms.

The paper is well written and the contributions are well presented.


**Q4 Main Weakness:**

The major issue in this paper is that it is missing comparative analysis. There are many related work. The authors do not compare to any of them, including the ones cited in the paper. We think that it is important to compare to existing path finding algorithms, including recent as well as conventional algorithms such as nudged elastic band, string method and its variations.


**Q5 Detailed Comments To The Authors:**

A major criticism is the missing comparative analysis in this work. The paper would be benefit from comparing the proposed method to other methods, by including comparison with normalizing flow methods on one hand, and existing path finding algorithms on the other hand. Even though the proposed method does not outperform well known and investigated conventional algorithms such as nudged elastic band and string method, it is important to assess how much we still have to do in order to outperform state-of-the-art methods.

We didn’t find major spelling or grammatical issues in the paper



**Q7 Justification For Your Score:**

Our overall assessment is based on all the aforementioned comments.


**Q9 Complying With Reviewing Instructions:**

1: Yes.

---

### Official Review · Reviewer_v1sN · 2022-04-15

**Q2(1) Originality/Novelty:** 3
**Q2(2) Significance/Impact:** 3
**Q2(3) Correctness/Technical Quality:** 3
**Q2(6) Clarity Of Writing:** 4
**Q6 Overall Score:** 7
**Q8 Confidence In Your Score:** 2

**Q1 Summary And Contributions:**

This submission is about learning representations of molecular dynamics that model the Boltzmann distribution while simultaneously making it easy to find minimum energy transition paths. It combines a normalizing flow model to approximate the distribution with a sophisticated algorithm for approximating the path cost. Experiments show the ability to approximately identify the transition state in cases where it's known.

**Q2 Assessment Of The Paper:**

More detailed information regarding each of these aspects is given below:

**Q2(4) Quality Of Experiments (Optional):**

3: Good: The experimental evaluation is adequate, and the results convincingly support the main claims.

**Q2(5) Reproducibility:**

3: Good: Key resources (e.g., proofs, code, data) are available and key details (e.g., proofs, experimental setup) are sufficiently well-described for competent researchers to confidently reproduce the main results.

**Q3 Main Strengths:**

As someone with no background at all in molecular simulation, I learned a lot from reading this paper. The writing is clear, especially given the difficulty of the topic, and all of the choices seem well motivated. Improvements to this problem would have significant practical impact. I have not spotted any red flags. I'm unable to judge the comprehensiveness of the related work, but I don't have any reason to doubt it.


**Q4 Main Weakness:**

The topic is fairly unusual for an ML conference (only a handful of the citations are from ML venues). This isn't a problem, since the task is interesting and important and it's likely there will be an audience in ML. It does make the paper hard for me to evaluate, and despite the overall very high quality of the writing, more can be done to make it accessible to an ML audience.

My main uncertainty from an evaluation standpoint is how crucial the ML component is. The density modeling and path costs are separate terms in the objective; the former is generic ML, and the latter seems like a sophisticated algorithm from molecular dynamics. What is gained from having the density model?  (I elaborate on this question below.) Is the path cost approximation a standard technique, or did it need to be adapted to accommodate the distribution modeling component?

Will software be released?  The algorithm seems complicated to implement.

**Q5 Detailed Comments To The Authors:**

I asked above, "what is gained from having the density model?". I imagine the advantage of training a flow model, rather than simply computing an MEP, is to amortize the MEP computation over multiple start/end pairs. But in the experiments section, it seems that the main object of interest is the transition point, which (if I understand right) is independent of the endpoints. Or is there some other advantage?

It would be useful to give a concrete example of collective variables.

I found the use of delta functions starting in Eqn. (6) a little confusing. I don't think these sorts of manipulations are common in ML.

The Hessian-vector products are estimated using finite differences. The Pearlmutter trick should be much more numerically stable and approximately the same computation cost. It's implemented in most modern DL frameworks, so you might as well use it.

Minor:

- I wasn't clear on the distinction between x and r in Section 3.1
- Is it an assumption that there are exactly two metastable states and these are known in advance?
- p. 4: "Eq. (3) is yet": should this be "not yet"?
- p. 4: "extremely high dimensionality"
- Did a factor of 1/S disappear in Eqn. (13)?  Also, what is I in this equation?



**Q7 Justification For Your Score:**

This submission is interesting, and I don't see any significant flaws. My score is limited by my not understanding the topic well enough to feel confident assigning a higher score. I may raise it based on the answers to my questions under "weaknesses".

**Q9 Complying With Reviewing Instructions:**

1: Yes.

---

### Decision · Program_Chairs · 2022-05-15

**Decision:**

Accept (Poster)

**Comment:**

Meta Review: This is a nice paper on generating Boltzmann samples and discovering transition paths for describing the transition mechanisms. The algorithm the authors propose, PathFlow functions not only as a one-shot sample generator but also a transition pathfinder. The authors validated their approach using synthetic experiments and a quasi-real world application.

Overall, the reviewers felt that this paper is worthy of publication with minor modifications.

The authors should address the following criticisms in their camera-ready version:
1. Make it more accessible to the broader machine learning community.
2. Include comparative analysis you described in your response to reviewer g8Zi.
3. Improve/Fix grammar/english.